# Percutaneous versus Transcutaneous Electrical Stimulation of the Posterior Tibial Nerve in Idiopathic Overactive Bladder Syndrome with Urinary Incontinence in Adults: A Systematic Review

**DOI:** 10.3390/healthcare9070879

**Published:** 2021-07-13

**Authors:** Aida Agost-González, Isabel Escobio-Prieto, Azahara M. Pareja-Leal, María Jesús Casuso-Holgado, María Blanco-Diaz, Manuel Albornoz-Cabello

**Affiliations:** 1Departamento de Fisioterapia, Facultad de Enfermería, Fisioterapia y Podología, Universidad de Sevilla, 41009 Sevilla, Spain; aidaagost97@gmail.com (A.A.-G.); mcasuso@us.es (M.J.C.-H.); malbornoz@us.es (M.A.-C.); 2Servicio Madrileño de Salud, D.A. Norte, 28035 Madrid, Spain; azaharapl@gmail.com; 3Departamento de Fisioterapia, Facultad de Medicina y Ciencias de la Salud, Universidad de Oviedo, 33006 Oviedo, Spain; blancomaria@uniovi.es

**Keywords:** percutaneous electric nerve stimulation, transcutaneous electric nerve stimulation, adult, urinary bladder, overactive, urinary incontinence, tibial nerve

## Abstract

Background: Percutaneous electrical stimulation and transcutaneous electrical stimulation (PTNS and TTNS) of the posterior tibial nerve are internationally recognized treatment methods that offer advantages in terms of treating patients with overactive bladder (OAB) who present with urinary incontinence (UI). This article aims to analyze the scientific evidence for the treatment of OAB with UI in adults using PTNS versus TTNS procedures in the posterior tibial nerve. Methods: A systematic review was conducted, between February and May 2021 in the Web of Science and Scopus databases, in accordance with the PRISMA recommendations. Results: The research identified 259 studies, 130 of which were selected and analyzed, with only 19 used according to the inclusion requirements established. The greatest effectiveness, in reducing UI and in other parameters of daily voiding and quality of life, was obtained by combining both techniques with other treatments, pharmacological treatments, or exercise. Conclusions: TTNS has advantages over PTNS as it is more comfortable for the patient even though there is equality of both therapies in the outcome variables. More research studies are necessary in order to obtain clear scientific evidence.

## 1. Introduction

The International Classification of Diseases (ICD-11) defines overactive bladder syndrome (OAB), with code GC50.0, as a urological condition characterized by voiding urgency, polyuria, and nocturia that may or may not be accompanied by urinary incontinence (UI) [1]. OAB presents a worldwide prevalence [2] of 16% to 23%, rising to 15% in those over the age of 40 years [3] and to 30–40% in those over 75 years [4], although it can be found in people of all ages. Its prevalence in Europe [5] is around 12%.

J.C. Angulo, in an article [6] published in 2016, revealed a 19.46% prevalence of OAB in the Spanish population, with at least one episode of urge UI (UUI) a day in 48.74% of cases [6]. This prevalence is greater in females than in males. Studies conducted [7] in the populations of Europe, the United States, Asia, and Africa reveal a prevalence of UUI of 1.5% to 14.3% in men aged between 18 and 20 years, whereas in women, it ranges from 1.6% to 22.8%. The same is true for those aged over 30 years, where the prevalence in men is from 1.7% to 13.3%, as opposed to 7% to 30.3% in women [7].

Among subjects with OAB on the global scale, UUI has been observed to be the most unpleasant symptom of this condition [4]. Further, people with OAB usually adopt certain coping strategies that involve a decrease in their quality of life and socialization, such as limiting liquid intake, avoiding traveling, and attempting to have direct access to toilets [3].

UI is the involuntary leakage of urine and lack of ability to control urination, accompanied by spontaneous contractions of the detrusor muscle. There are various subtypes of UI: urgency UI (UII), the sudden desire or need to urinate; stress UI (SUI), caused by efforts, physical exercise, sneezing, or coughing; mixed UI (MUI), combined with urgency and efforts [1] (code MF50.2 in ICD-11). To be able to determine the best treatment option in each patient, a personalized assessment is necessary, including the evaluation of different aspects of health, motivation, and availability or access to specific treatments [2].

Profitability is a fundamental aspect when it comes to reviewing treatment options in this type of condition, which entail great social and financial costs. Previous studies [7] showed a value of EUR 7 billion in subjects with OAB over 18 years old in Canada and European countries, including Spain [7].

There are several alternatives for OAB and UI treatment: behavioral treatments, considered first-line treatments; pharmacological or second-line treatments such as anticholinergic or antimuscarinic and b-adrenergic drugs, and, by way of a third line of treatment, injections of OnabotulinumtoxinA and therapies with electrical stimulation, including, among others, percutaneous and transcutaneous electrical stimulation (PTNS and TTNS, respectively), which are the object of this study [8].

With regard to treatment by electrical stimulation of the posterior tibial nerve (PTN), this involves retrograde stimulation of the nerve fibers of the sacral plexus, which innervates the bladder and detrusor muscle [2,3,4,5,8]. Electrical stimulation can be applied through insertion of a needle in the PTN—that is, PTNS is carried out in the said nerve—or through surface electrodes, with TTNS [9], with beneficial and safe effects in the short term in women with OAB, and no relevant adverse effects [10], according to the review by Sousa-Fraguas et al., 2020.

These techniques may represent an advantage in treatment of subjects with OAB who present UI, enabling these difficulties to be solved, as they can be compared favorably to treatment using antimuscarinic drugs, due to them being less costly [11].

In this respect, the present study aimed to summarize the knowledge available and conduct a critical analysis of the evidence from randomized controlled clinical trials, observational studies, systematic reviews, and meta-analyses on the effectiveness of PTNS and TTNS in the treatment of adults with OAB who present UI.

## 2. Materials and Methods

This systematic review followed the Preferred Reporting Items for Systematic Reviews and Meta-Analyses (PRISMA) guidelines [12]. The protocol was registered in the *International Prospective Register of Systematic Reviews* (PROSPERO/NHS)—number: 184809.

### 2.1. Selection Criteria

Three researchers independently reviewed the articles found. In order to formulate the objective and the question of the review, the PICOS strategy was used [13] (P—population or patients; I—intervention; C—comparison; O—outcomes; S—study design), in which P = (adults with OAB syndrome (OABS) and presence of UI); I = (PTNS and TTNS); C = (control group that received no intervention or received standard/usual care); O = (randomized clinical trials (RCTs), descriptive, observational studies, systematic reviews, and meta-analyses), and S = (randomized controlled clinical trials, descriptive observational studies, systematic reviews, and meta-analyses). This strategy enabled the establishment of critical reasoning on the issue [13] and the formulation of the following question: “What is the existing scientific evidence on the treatment of adults diagnosed as having OABS with UI through procedures of PTNS versus TTNS?”.

### 2.2. Data Sources

The bibliographic search was performed between the months of February and May 2021. The search terms used were percutaneous electric nerve stimulation; transcutaneous electric nerve stimulation; adult; urinary bladder, overactive; urinary incontinence; tibial nerve. Two multidisciplinary databases, Scopus and Web of Science (WOS), were used in the search. The search strategy followed is presented in Table 1.

### 2.3. Data Collection and Analysis

By way of exclusion criteria: all articles not published in English or Spanish; studies conducted in patients with neurological diseases or UI exclusively of neurogenic origin; carried out in children, animals, or patients with an associated underlying pathology; addressing fecal incontinence; in which treatment was not carried out with PTNS or TTNS of the PTN, or not aimed at treatment of OAB with UI; narrative or nonsystematic reviews; all documents not aligned with the research problem. The bibliographic research focused on all articles published from 2015 to 2020.

In order to obtain reliable, valid results, without them being influenced by bias, the Physiotherapy Evidence Database scale (PEDro) [14] was used to assess the methodological quality of the experimental studies, based on the Delphi list [15]. In the same way, the STROBE declaration [16] was applied for the evaluation of observational-type studies, and the PRISMA declaration [14] for reviews that followed its criteria in their execution. Articles that did not exceed the score of five in the PEDro scale [14] or with a score of less than 11 points in the STROBE declaration [16] were excluded, finally obtaining the articles chosen for the review.

## 3. Results

### 3.1. Literature Search

Figure 1 shows the PRISMA flow chart of this systematic review.

The initial search in the databases gathered a total of 259 articles, 98 from Web of Science (WOS) and 161 from Scopus.

The initial screening phase produced 130 articles after removing duplicates (*n* = 129).

Based on the titles and abstracts of the articles, a total of 56 articles were removed. Then, considering the remaining 74 eligible studies, many were excluded after full-text reading (*n* = 47), or because it was not possible to access the full text (*n* = 3), or for not passing the methodological quality scale (*n* = 5).

Finally, 19 studies [3,4,5,17,18,19,20,21,22,23,24,25,26,27,28,29,30,31,32] were included. Of these, nine were experimental [3,17,18,19,20,21,22,23,24], including eight RTCs [3,17,18,19,20,21,22,24] (Table 2); four were observational studies [5,25,26,27] (Table 3), and six were either systematic reviews [4,29,31] or a meta-analysis [28], while two encompassed both types of studies [30,32] (Table 4).

### 3.2. Summary of the Evidence

As for the comparison between PTNS and TTNS therapy, studies were found [3,17] in which significant changes were observed in the variables of diurnal frequency of urination, nocturnal frequency of urination, 24-h voiding frequency, mean voided volume, and number of episodes of UI and UUI in 24 h [17].

When TTNS combined with trospium chloride [20] was compared to placebo, a decrease in frequency of urination was observed in both groups (*p* = 0.001 and *p*= 0.003, respectively); as to mean voided volume, significant improvements were observed in both groups, with greater significance in the combined therapy (*p* = 0.005), although there was a significant delay in the combined therapy group with regard to the first sensation of a full bladder [20].

In other studies [23,32], upon combining PTNS with drugs, 35/53 participants completed the satisfaction survey after treatment, 66% of whom preferred to continue with maintenance treatment, with a mean interval of 44.4 days (7–155 days) and frequency of sessions of 1.1 months; attendance was observed if there were symptoms of OAB, while patients with multiple sclerosis had the possibility of returning [23].

Following this comparative line, one review was found [4] in which TTNS was compared with diverse therapies, including 3/10 studies that compared simulated therapy, 4/10 anticholinergic, 1/10 exercise, 1/10 behavioral, and 1/10 two different stimulation sites. The three remaining studies compared TTNS with other treatments: extended-release oxybutynin vs. TTNS + fármacoM; TTNS vs. transcutaneous sacral foramina vs. combination of the two; bladder and pelvic floor training vs. TTNS.

By contrasting daily or weekly treatment [21] with TTNS, 100% of weekly participants completed the compliance and experience questionnaire, in comparison to 90.5% of patients on daily therapy [21]. Although 53% (18) gave as a result a moderate or significant improvement in symptoms for the global response assessment (GRA), 75% (13/20) of neurological patients with OAB and 36% (5/14) of patients with idiopathic OAB responded to the intervention [21].

With respect to adherence to treatment [5,17] with TTNS, one of the studies [5] established different reasons for discontinuity (in 70 participants): lack of symptom relief (70%); difficulty in complying (6%); becoming asymptomatic (8%). However, 16.9% (14) of patients continued treatment, with a mean follow-up of 39.3 months [5].

Meanwhile, a BMI of obesity (=30 kg/m^2^) was observed to be the only statistically significant variable predictive of failure in the response to PTNS (*p* = 0.002) [27]. Notably, after PTNS therapy, 66% (19/29) of participants informed of an improvement in their symptoms [27].

Some of the additional complications to those observed in the analysis of results [3,19,21,23,26,28,30,32] were urinary tract infections in 10/17 studies (peer comparisons revealed that OnabotulinumtoxinA was associated with a greater incidence of urine infections vs. placebo, sacral neurostimulation (SNS), and PTNS); ranking in order of fewest infections: first PTNS, second SNS, third placebo, and fourth OnabotulinumtoxinA. Further, urine retention with a need for intermittent catheterization was found in 11/17, with peer comparisons showing that OnabotulinumtoxinA was associated with a greater incidence of retention vs. placebo, SNS, and PTNS; ranking in order of lowest incidence: first SNM, second placebo, third PTNS, and fourth OnabotulinumtoxinA [30].

## 4. Discussion

The aim of this systematic review was to analyze the scientific evidence on the treatment of OAB with UI through procedures of PTNS, compared to TTNS, of the PTN. Nineteen studies were included, which analyze, observe, and compare these therapies with other methods, such as simulated treatment, placebo, anticholinergic or other drugs, sacral electrical stimulation, or vaginal electrical stimulation.

Among the studies whose intervention was based mainly on PTNS or TTNS therapy vs. another therapy, UI presented significant improvement when compared to placebo or simulated treatment [28,29,31]. Abulseoud A et al. [20] showed a significant improvement in the number of episodes of UI in combined groups of TTNS and trospium choloride compared to TTNS for eight weeks [20]. It is worth noting the significant improvements observed in the review by Veeratterapillay R et al. [31] in UUI after 12 weeks of treatment and two years of maintenance with PTNS therapy, unlike what was observed by Welk B et al. [22] in their RCT with PTNS therapy, with no significant differences between TTNS treatment compared to simulated therapy.

It is worth highlighting the improvements in UI reflected in the systematic review conducted by Booth J et al. [4], when combining TTNS therapy with pelvic floor exercises or behavioral treatment, as well as the results observed in the systematic review and meta-analysis performed by Wang M et al., as regards the reduction in the number of episodes of UI and UUI per day through PTNS therapy [32].

Apart from two studies analyzed [5,24], parameters related to frequency of urination, urgency of urination, and nocturia, as well as other symptoms of OAB such as voiding volume and urodynamic changes were included as variables and presented dissimilar results between studies.

In a recent study [10] from 2020, significant improvements were observed in the perception of quality of life of patients treated with TTNS and PTNS, with no differences between treatments [10].

By focusing on the quality of life observed in the studies analyzed, it is worth highlighting that all the experimental studies included provided data on this. Some of these studies [17,18,21,23] showed significant improvements in quality of life, through diverse questionnaires, after treatment with PTNS or TTNS, revealing that this improvement increased when TTNS was combined with trospium chloride, although the difference was not significant [20].

In 2013, Peters KM et al. [11] observed improvements in the quality of life of patients with OAB who were treated with PTNS, evaluated three years after treatment. In the present review, PTNS has been seen to present significant differences in quality of life when compared to vaginal electrostimulation [18], and it has been observed that there are significant differences in increased quality of life in both neurogenic and non-neurogenic OAB [23].

As for other parameters, it is worth noting the RCT of de Scaldazza CV et al. [18] in which significant differences were revealed in terms of the patient’s global perception in favor of the PTNS technique compared to vaginal electrostimulation.

Leroux PA et al. [5] in their study showed some of the reasons why there is discontinuity in treatment with TTNS therapy, the most prevalent of which, in 70% of cases, was sufficient relief from symptoms, while in 6% it was due to complications for compliance with the treatment, and in 8% it was due to a complete reduction of the symptoms and becoming asymptomatic [5].

Most of the studies included in this review report of the absence of adverse effects during treatment [5,17,18,20,22,24,25]. Studies that combine PTNS and TTNS therapy [17], and those in which TTNS therapy is involved [5,20,22,25], point out that there are no adverse effects after the use of this therapy, except for the study conducted by Moratalla-Charcos LM et al. [26], who speak of mild pain on plantar flexion after the use of this technique.

Regarding PTNS therapy, no serious adverse effects were found, only minor bleeding episodes were mentioned or mild discomfort at the needle insertion site [3,25,29], sometimes causing hematomas or paresthesia at the puncture site [31].

As for the electrical stimulation parameters with PTNS therapy, most studies referred to weekly sessions for 12 weeks as the time of treatment. Although some of them did not show the other parameters, the rest coincided with regard to sessions of 30 min duration, a frequency of 20 Hz, pulse of 200 ms, 34-gauge needle inserted approximately five cm above the medial malleolus, and electrode in ipsilateral calcaneus [17,19,22,23,27]. In terms of amplitude, this was increased to the level of discomfort of the patient, feeling of tickling on the sole of the foot, or flexion of the big toe.

Upon referring to treatment with TTNS therapy, the parameters between the studies are more variable: some of the studies mentioned the same stimulation parameters as those of PTNS therapy, while others varied in terms of frequency, using a frequency of 10 Hz, as was the case of the randomized clinical trial of Welk B et al. [22], also highlighting the frequency of weekly sessions, with a total of three weekly sessions for 12 weeks.

The experimental studies of Abulseoud A et al. [20] and Seth JH et al. [21] stand out due to the use of different parameters, with the former [20] using frequencies of 10 Hz, pulse of 250 ms, treatment three times a week for eight weeks, and with a stimulation time of 30 min. Meanwhile, in the latter study [21], they used amplitudes of 27 mA, pulse between 70 and 560 ms, which varied depending on patient tolerability, for 12 weeks, both daily and weekly. Mallman S et al. [24], in their RCT, showed a stimulus duration of 20 min with a follow-up of six weeks and a pulse duration of 300 ms.

## 5. Conclusions

It is complicated to be able to establish which electrical stimulation therapy of the PTN is the most effective for treatments of idiopathic OAB with UI in adults, as far as the different parameters observed in this review are concerned, due to the variability of the results obtained and the electrical stimulation parameters used in the studies included. Nevertheless, it is worth highlighting the advantages TTNS therapy presents with respect to PTNS therapy, as this could be more comfortable for the patient, all things being equal in the results variable.

## Figures and Tables

**Figure 1 healthcare-09-00879-f001:**
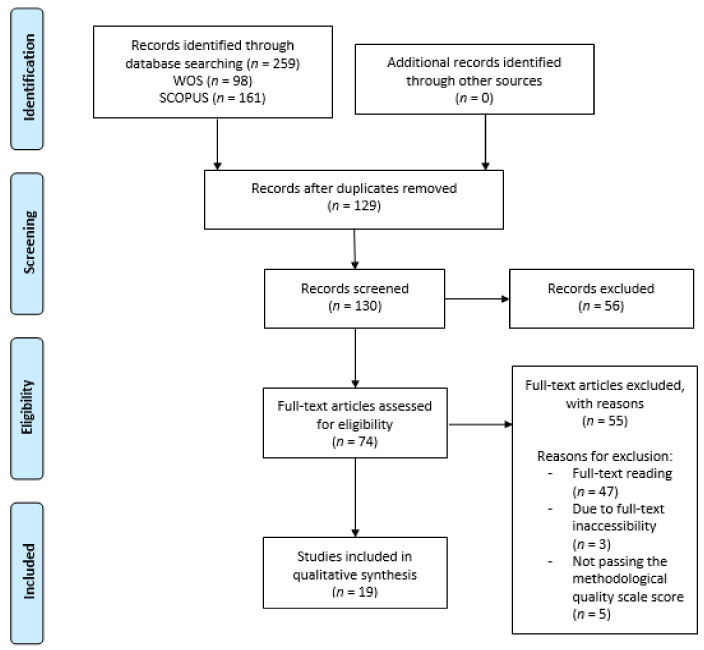
Flow chart of this systematic review.

**Table 1 healthcare-09-00879-t001:** Search strategy in WOS and Scopus databases.

Databases	Search Strategy
WOS (February–March 2021) (years 2015/2020)	I. (“Transcutaneous Electric Nerve Stimulation” OR “Therapy, Percutaneous Neuromodulation” OR “Electrical Neuromodulation, Percutaneous”) AND adult AND (“Urinary Incontinence” OR “Urinary Bladder, Overactive”) AND (“tibial nerve” OR “Posterior tibial nerve”). II. (“Transcutaneous electrical nerve stimulation” OR “Transcutaneous tibial nerve stimulation” OR “transcutaneous stimulation tibial nerve”) AND adult AND (“overactive bladder” OR “detrusor activity” OR “urinary incontinence”). III. (“percutaneous tibial nerve stimulation” OR “PTNS”) AND adult AND (“overactive bladder” OR “detrusor activity” OR “urinary incontinence”).
Scopus (February–March 2021) (years 2015/2020)	I. (“Transcutaneous Electric Nerve Stimulation” OR “Therapy, Percutaneous Neuromodulation” OR “Electrical Neuromodulation, Percutaneous”) AND adult AND (“Urinary Incontinence” OR “Urinary Bladder, Overactive”) AND (“tibial nerve” OR “Posterior tibial nerve”). II. (“Transcutaneous electrical nerve stimulation” OR “Transcutaneous tibial nerve stimulation” OR “transcutaneous stimulation tibial nerve”) AND adult AND (“overactive bladder” OR “detrusor activity” OR “urinary incontinence”). III. (“percutaneous tibial nerve stimulation” OR “PTNS”) AND adult AND (“overactive bladder” OR “detrusor activity” OR “urinary incontinence”).

**Table 2 healthcare-09-00879-t002:** Characteristics of experimental studies included in the systematic review. Sevilla, ES. 2021.

**Author/s**	**Study Design**	**Study Population**	**Sample Size**	**Intervention**	**Follow-Up**	**Randomization**
Ramírez-García I., et al., 2019 [17]	RCT	OAB and OD	*n* = 68 (46 W and 22 M) (34 per G) Mean age (MA): 59.6 ± 27.4 kg. Symptom duration: 1–56 years with mean of 5.1.	Grupo A(GA) (intervention)→ TTNS Grupo B (GB) (control)→ PTNS	12 weeks	Online 1:1 Randomization Sequence“Sealed Envelope Lt. 2015”
Scaldazza C.V., et al., 2017 [18]	RCT	W with OAB	*n* = 60 W (30 per G) MA: 58.5 years (38–72)	GA→ EMS vaginal + exercises GB→ PTNS	1 m post-treatment.	Online randomization GraphPad QuickCalcs software
Preyer O., et al., 2015 [19]	RCT Pilot study	W with OAB, with no previous PTNS or anticholinergic treatment	*n* = 36 W (18 per G) GA: MA = 57.4 ± 9.5 GB: MA = 55.8 ± 16.2	GA→PTNS GB→ Tolterodine	3 m (baseline/1–3 m)	1:1 adapted randomization method (computer assistant)
Abulseoud A., et al., 2018 [20]	RCT	W with OAB and OD, and behavioral treatment failure	*n* = 30 W (15 per G) G 1: MA = 48 ± 16.42 G 2: MA = 48.13 ± 10.80	G 1: TTNS + placebo bottle identical to G 2. G 2: TTNS + Trospium Chloride	8 Weeks	1:1 by random number table
Martín-García M. & Cramptom J. 2019 [3]	RCT	W with non-neurogenic OAB who responded to PTNS initial treatment of 12 weeks	*n* = 24 W (12 per G) G PTNS: MA = 58 ± 10 G TTNS: MA = 54 ± 12	Maintenance G PTNS G TTNS	6 m Evaluation: baseline (post-treatment with 12 weeks PTNS), at 6 weeks, 3 m and 6 m	1:1 via sealed, opaque envelopes with numbered sequences
Seth J.H., et al., 2018 [21]	Randomized pilot trial	M/W with OAB after treatment with ineffective conservative therapy	*n* = 48 (24 multiple sclerosis and 24 Idiopathic OAB) G 1 (daily treatment): MA = 46.4(32–73), 18 W/6 M, 20 UI G 2 (weekly treatment): MA = 46.9(20–81), 20 W/4 M, 18 UI	TTNS G 1: 1 session/day G 2: 1 session/week	12 weeks, with evaluation at 4, 8 and 12 weeks	Stratified method by sealed envelopes
Welk B., et al., 2020 [22]	RCT	W with OAB, neurological diseases with urinary urgency with or without UI	*n* = 50. MA (73%, 22/30); 10% (3/30) used spontaneous voiding catheter G TTNS→ *n* = 26; MA = 62 (54–68) GS → *n* = 24; MA = 53 (46–64)	G TTNS→ TTNS. Increasing amplitude to maximum tolerance or flexion of the big toe GS→ TTNS. Constant amplitude	12 weeks	1:1 by random number generator
Tudor K.I., et al.020 [23]	Experimental study of two groups related retrospective	Neurological or idiopathic OAB refractory to 1st line treatments	*n* = 74 (52 W/22 M), MA = 56.0 (25.2, 59.8). 49 (66.2%)→ neurogenic OAB 25 (33.8%) → idiopathic OAB	PTNS + drug	12 weeks	Without randomization
Mallman S., et al., 2020 [24]	RCT	W with OAB	*n* = 50 (25 per G) MA = 61.48 ± 10.10	G transcutaneous sacral EMS G TTNS	6 weeks	Sequence generated in 2 G by WinPEPI version 11.63
**Author/s**	**Variables**	**Results**		**Conclusions**	**Adverse Effects and Limitations**	**Met** **Quality**
Ramírez-García I., et al., 2019 [17]	Difference day and night urination Mean voiding volume Number of urgencies and UI Quality of life	At the beginning/12 weeks. GA: *n* = 34/GB: *n* = 34 Differences by protocol: GA: *n* = 32 /GB: *n* = 29. Post intervention differences: GA: *n* = 34/GB: *n* = 34. Difference adjustment with 95% confidence interval Quality of life I-QoL: GA (21.5 p), GB (22.1 p) (*P* < 0.001)		Both techniques improve symptoms and quality of life Treatment adherence (*P* = 0.236).	No serious adverse effects Study carried out to measure differences between groups. High cost.	7/10
Scaldazza C.V., et al., 2017 [18]	Number of voiding in 24 h Number of UUI episodes Nocturia Quality of life (OAB-q SF) Bladder urgency perception (PPIU-S) Perception of global improvement (PGI-I)	Number of daily voiding → GA→ *P* = 0.0620; GB→ *P* = 0.0307; Difference between groups (DG)→ *P* = 0.3758 Number of UUI episodes →GA: *P* = 0.1293; GB→ *P* = 0.0009; DG: *P* = 0.0251 Nocturia→ GA: *P* = 0.1683; GB→ *P* = 0.0201; DG: *P* = 0.049 Voiding volume→ GA: *P* = 0.0048; GB→ *P* = 0.0003; DG: *P* = 0.0222 OAB-q SF: 6 items: GA: *P* = 0.0420; GB: *P* < 0.0001; DG: *P* = 0.0172. 13 items: GA: *P* = 0.0420; GB: *P* < 0.0001; DG: *P* = 0.0295 PPIU-S: GA: *P* = 0.1014; GB: *P* = 0.0001; DG *P* = 0.0459 PGI-I→ GA: *P* = 0.0415; GB: *P* = 0.0415		PTNS 1st line treatment (efficacy, minimally invasive)	Without significant side effects	6/10
Preyer O., et al., 2015 [19]	Differences in number of voids in 24 h between both groups (voiding diaries). Number of UI episodes Quality of life (QoL-VAS)	*n* = 16 per group Number of voids→ without significant decrease 1–3 m in both G (*P* = 0.13), DGNS (*P* = 0.96). No significant differences in number at the beginning and post treatment (*P* = 0.79) Quality of life (QoL-VAS)→ depends on the initial values, in GB mean values are lower than GA in 1–3 m Increase in both G 1 and 3 m, without significant changes (*P* = 0.07) Number of UI episodes in 24 hours→ depends on the number episodes at the beginning of treatment (*P* = 0.0001). NSDG pre/post treatment (*P* = 0.89). Significant changes in 3 m (baseline/1 m) (*P* = 0.03)		Both are effective treatments: decrease in number of UI episodes, but not in urinary frequency. PTNS has fewer side effects.	The first 4 weeks mainly GB→ dry mouth and dizziness. 9 participants (3 m) GA→ pain in puncture area. 3 participants (3 m). Small sample size and standard deviations greater than expected, which could be due to a type II error; no blinding	6/10
Abulseoud A., et al., 2018 [20]	Brief OAB Symptom Score (OABSS) Short form of incontinence impact questionnaire (IIQ-7)	OABSS → post-treatment G (*P* < 0.001) and G 2 (*P* = 0.024) IIQ-7 → (G1: *P* = 0.002; G 2: *P* = 0.001). Pre- treatment min 50 points (good quality of life), post-treatment 20 (6 G 1 and 14 G 2). Pre- treatment→ severe OAB symptoms in 26 patients (12 G 1 and 14 G 2), post-treatment 4 (G 1). Cystometric volume → post-treatment (G1: *P* = 0.026; G 2: *P* = 0.001), G 2 (*P* = 0.034)		TTNS is tolerable and effective when combined with Trospium Chloride. Better results without side effects.	Without side effects Limitation: need for another group with sodium chloride treatment without TTNS. Longer follow-up studies needed.	8/10
Martín-García M. & Cramptom J. 2019 [3]	Urinary frequency Number of urgency episodes Number of UUI episodes Severity of symptoms Quality of life (HRQoL)	Urinary frequency G TTNS→ decreasing from the beginning/6 m: 8.5 (1.9) vs 7.7 (2.8), (*P* = 0.373).; G PTNS→ during the study (*P* = 0.242), increasing from 7.3 (4.7) to 8.7 (2.4) at 6 m *P* = 0.208. Number of UUI episodes in 24 h: G TTNS→ (*P* = 0.900); G PTNS→ (*P* = 0.655) Number of urgency episodes in 24 h: G TTNS→ Friedman test (*P* = 0.038). Wilconxon test: significant increase baseline/ 6 Weeks: 1.7 (2.8) vs 3.2 (3.6), (*P* = 0.044); and baseline/3 m: 1.7 (2.8) vs 2.5 (2.6 (*P* = 0.011). Without significant changes baseline/6 m: 1.7 (2.8) vs 2.0 (1.4) (*P* = 0.325); G PTNS→ (*P* = 0.883) Severity of symptoms → G TTNS→ (*P* = 0.584); G PTNS→ (*P* = 0.854) Quality of life HRQoL→ G TTNS→ (*P* = 0.676); G PTNS→ (*P* = 0.948)		Application of bilateral TTNS is an effective and tolerable treatment for the maintenance of benefits in OAB symptoms with previous PTNS therapy For maintenance: PTNS, regular visit for consultation; Home TTNS	TTNS without side effects. PTNS → 3 minor episodes of needle insertion bleeding (2 participants); 1 episode of discomfort/ pain over the needle area	7/10
Seth J.H., et al., 2018 [21]	Quality of life (ICIQ-OAB) and (ICIQ-LUTqol) Part A for severity of symptoms and Part B for patient discomfort. 3-day voiding diary → urinary frequency in 24 h; number of UI episodes.	Quality of life → improvements in ICIQ-OAB score and ICIQ-LUTSqol score between the beginning and during 12 week treatments in both G. Daily treatment → part A - ICIQ-OAB → means improved between the beginning and 12 weeks from 9.3 (2.5) to 7.5 (3.1) Part B — ICIQ-OAB → from 29.6 (8.1) to 25.6 (9.5) Part A — ICIQ-LUTSqol→ from 51 (12.8) to 44.2 (13.1) Part B — ICIQ-LUTSqol → from 130.3 (43.7) to 105.5 (57.8) Weekly treatment→ part A - ICIQ-OAB→ week 12 from 9.1 (1.9) to 5.9 (1.7) Part B — ICIQ-OAB → from 29.7 (5.9) to 19.1 (8.5) Part A — ICIQ-LUTSqol → from 44.9 (9.0) to 35.9 (8.8) Part B — ICIQ-LUTSqol→ from 102.1 (40.1) to 63.9 (42.8) Urinary frequency in 24 h → Daily treatment → from 10.8 to 8.2 at 12 weeks Weekly treatment→ from 12.2 to 9.5 at 12 weeks Number of UI episodes → Daily treatment → from 2.8 to 1.6 at 12 weeks. Weekly treatment → from 2.3 to 0.9 at 12 weeks		Safe treatment. Low frequency stimulation (1 Hz) improves quality of life and symptoms of voiding diaries (daily/weekly) Neurological patients respond more frequently to treatment (65%) versus patients with idiopathic OAB (36%). No significant differences in tolerability between G	No significant complications during treatment or in satisfaction surveys 5 patients said therapy was uncomfortable and did not continue the treatment No significant safety problems. One patient developed skin redness in the area of stimulation Limitations: high dropout rate (in relation to device, lack of improvement, or local discomfort)	5/10
Welk B., et al., 2020 [22]	Questionnaire (PPBC) Compress weight in 24 h, for UI 3-day voiding diary→ Urinary frequency and functional capacity in 24 h Quality of life (OAB-q SF, in G with OAB) Neurological patients → (NBSS), (Qualiveen-SF)	PPBC→ 13% (3/24) of sham patients and 15 (4/26) of TTNS treatment were considered responders (*P* = 0.77) Marginal mean of the end of the PPBC score was 3.3 (2.8–3.7) for TTNS vs 2.9 (2.5–3.4) for simulated (*P* = 0.30) Compress weight in 24 h→ NSDG (*P* = 0.64) Functional capacity → NSDG(*P* = 0.12) Urinary frequency in 24 h → NSDG(*P* = 0.32) OAB-qSF Questionnaire → NSDG in symptom discomfort (*P* = 0.82) and quality of life (*P* = 0.29) NBSS→ NSDG (*P* = 0.16) Qualiveen-SF→ NSDG (*P* = 0.85) Global assessment of improvement → NSDG (*P* = 0.27)		TTNS does not display greater efficacy in patient perception of OAB symptoms and objective parameters evaluated	With no adverse effects during the study Limitation in results generality, most had UI and had failed with pharmacological therapy. Small sample	8/10
Tudor K.I., et al., 2020 [23]	ICIQ-OAB questionnaire and ICIQ-LUTSqol 3-day voiding diary → urgency and severity of UI	64 (86%) completed 12 weeks. Significant improvements at 12 weeks of treatment in ICIQ-OAB ICIQ-LUTSqol, change in urinary frequency over 24 h and severity of UI in bladder diary G neurogenic VH→ ICIQ-OAB (*P* = 0.04); ICIQ-LUTSqol (*P* = 0.05) [in ICIQ-OAB, odds ratio (IC 95%) 0,93 (0,87, 0,99), *P* = 0,03], severity of UI [in bladder diary, odds ratio (IC 95%) 0.05 (0.01, 0.63), *P* = 0.02] and QoL [IUTQ-LUTSqol, odds ratio (IC 95%) 0.98 (0.96, 0.99), *P* = 0.007] at 12 weeks		PTNS is a possible alternative treatment in patients with neurological disease and with ineffective or intolerable 1st line treatment	No adverse effects. 5 patients had mild discomfort at the needle insertion area Lack of blinding, lack of a placebo or control group, and lack of urodynamic assessment before treatment Not validated questionnaires in patients with neurogenic OAB	5/10
Mallman S., et al., 2020 [24]	Quality of life: KHQ Severity of UI: ISI Discomfort due to OAB symptoms: OAB-V8	NSDG (*P* > 0.005) OAB-V8: (6 weeks *P* = 0.0019) G TPNS/G transcutaneous sacral EMS KHQ e ISI: NSDG		Both therapies are effective and safe for the treatment of women with OAB, UUI, and MUI	No side effects	6/10

W = women; M = men; OAB = Overactive bladder syndrome; OV = overactive detrusor; MA = mean age s; G = Group; PTNS = Percutaneous electrical stimulation of the posterior tibial nerve; TTNS = Transcutaneous electrical stimulation of the posterior tibial nerve; m = month; EMS = electrical stimulation; BMI = body mass index; UI = Urinary incontinence; PP = Per protocol; PIT = Per Intention to treat; DG = difference between groups; I- QoL = Urinary Incontinence Quality of Life Scale ;UUI = urgency urinary incontinence; OAB-qSF = Overactive Bladder Questionnaire Short Form; PPIU-S = Scale Patient Perception on Intensity of Urgency Scale; PGI-I = Patient Global Impressions Scale or Disease Improvement; QoL-VAS = Qol. Visual Analogue Scale; OABSS = Overactive Bladder Screening Scale; IIQ7 = Incontinence Impact Questionnaire-7; HRQoL = Health Related Quality of Life; ICIQ-OAB = International Consultation on Incontinence Questionnaire Overactive Bladder Module; ICIQ- LUT = International Consults on Incontinence Lower Urinary Tract Symptoms; ICIQ-LUTSqol = International Consults on Incontinence Lower Urinary Tract Symptoms Quality of Life; PPBC = Patient Perception of Bladder Condition; NBSS = Neurogenic Bladder Symptom Score; Qualiveen-SF = Short Form of Lower Urinary Tract Dysfunction Impact on Quality of Life; NSDG = non-significant difference between groups; IC = confidence interval

**Table 3 healthcare-09-00879-t003:** Characteristics of the observational studies included in the systematic review. Sevilla, ES. 2021.

**Author/s**	**Study Design**	**Study Population**	***Sample Size***	**Intervention**		**Follow-Up**
Salatzki J., et al., 2019 [25]	Cross-sectional (cohort)	Positive response to PTNS treatments (10–12 weeks)	*n* = 83 PTNS-SEQ G 1: *n* = 28 G 2: *n* = 24 G 3: *n* = 31	G 1→ non-responders; no maintenance therapy G 2→ responders; possibility of maintenance therapy: they did not do it G 3→ responders who underwent maintenance therapy		18 weeks
Leroux P.A., et al., 2018 [5]	Prospective	Idiopathic or refractory OAB to anticholinergic treatment	*n* = 97 treated with TTNS 20 (21%) M; 77 (79%) W MA = 58.4 ± 16.6	**TTNS**		24 m
Moratalla-Charcos L.M., et al., 2018 [26]	Pilot study Prospective	OAB with or without OD/UI, without success in pharmacological treatment or dropout due to adverse effects	*n* = 45: 38 W and 7 M. MA = 66.6 ± 10.5 (41–83). OD: 53.3%.	**TTNS**		12 weeks
Palmer C., et al 2019 [27]	Retrospective	>65 with idiopathic OAB, after treatment with PTNS	*n* = 52: 23 M (44.3%); 29 W (55.8%). MA = 75.75 (65 to 93); BMI = 26.33 (17.4 to 43.9) kg/m^2^	**PTNS**		12 weeks
**Author/s**	**Variables**	**Results**		**Conclusions**	**Adverse Effects and Limitations**	**Met** **Quality**
Salatzki J., et al., 2019 [25]	ICIQ-OAB→ UI in OAB ICIQ-LUT→ OAB symptoms. 3-day voiding diary PTNS-SEQ questionnaire→ variables observation to return to maintenance	Groups 2 and 3→ improvements compared to G 1. Patients with idiopathic or non-neurogenic OAB → significant improvement vs neurogenic OAB (*P* = 0.048). Group 3→ return to treatment after 39–204 days; significant improvements in nocturia (ICIQ-LUT, *P* = 0.036) and voiding diary (*P* = 0.046) To identify variables back to maintenance, (nocturia in 3 days/daytime urinary frequency/ number of UUI episodes), ICIQ- OAB, ICIQ-LUT→successful to distinguish between G 2 and 3 (Chi-squared 11.23, *P* = 0.047) PTNS-SEQ(*P* = 0.039)→ + 75% of cases. Increase in the categories “lack of treatment effect” — greater probability of belonging to G 2. Alternatives found to treatment of PTNS in PTNS-SEQ: (PTNS *n* = 28; PTNS home *n* = 20; PTNS in medicine clinic *n* = 20)		12 weeks of PTNS→ safe and effective treatment for OAB. A beneficial response with PTNS in nocturia was a factor to return to maintenance. The voiding diary offers more objective results for the evaluation of the treatment	No side effects Limitations: small sample size; difference in number of participants between G; results only applicable to public health.	21/22
Leroux P.A., et al., 2018 [5]	Questionnaire effectiveness USP and USP-OAB Treatment discontinuity and adverse effects Comorbidities Drugs during treatment follow-up	3 (3%) died of unknown cause/10 were lost at follow-up TTNS persistence and predictive factors → mean follow-up = 39.3 (25–65) m; mean persistence TTNS 8.3 (1–40 m). persistence = 12 m/28 patients (29%) e = 18 m/16 patients (16%) Discontinuity risk factors → At 3 m = 24 (28,9%) abandonment TTNS. Baseline score USP-OAB > 11 predictor of early treatment failure (*P* = 0.014) (univariate analysis)		TTNS treatment for refractory OAB. Few patients continued long-term therapy, probably due to a decrease in efficacy over time.	No adverse effects or pain Limitation: loss in follow-up during the study, and lack of placebo group, and objective urodynamic data	17/22
Moratalla-Charcos L.M., et al., 2018 [26]	3-day voiding diary → Urinary frequency, nocturia, number of urgency episodes, number of UUI episodes, maximum voiding volume OABQ-SF Subjective improvements Satisfaction level	*n* = 39/45(86.6%)→ completed 12 weeks treatment Significant differences before and after treatment (*P* < 0.05) in urinary frequency, nocturia, number of urgency episodes, number of UUI episodes, maximum voiding volume. OABq-SF: *P* > 0.05. Statistically significant differences in GOD vs OAB Treatment satisfaction→ patients with mellitus diabetes (*P* = 0.043), in diabetes W (*P* = 0.042). In ordinal regression with 4 independent variables: number of vaginal deliveries (*P* = 0.011); psychiatric history (*P* = 0.001) were significant. Group with OD→ better satisfaction by increasing the number of vaginal deliveries and lower satisfaction for W and patients with diabetes. In ordinal regression with 3 independent variables: number of vaginal deliveries (*P* = 0.05) was significant		OAB treatment with TTNS is an effective, safe, minimally invasive and well tolerated therapy. In this study, all variables improved significantly compared to baseline.	Adverse effect: mild pain on plantar flexion, but no cases of dermatitis Limitation: lack of completion of voiding diaries, the OABQ-SF questionnaire, and subjective improvements, in addition to lack of a control group	17/22
Palmer C., et al 2019 [27]	OAB-V8 → OAB symptoms 3-day voiding diary Global impression of patient satisfaction (GIPS)	*n* = 21 (39%) used combination therapy during PTNS After PTNS→ 37 patients (70%) reported symptom improvements; 7 used anticholinergic, 6 used ß3 adrenoceptor agonist, 5 received intravesical injections of onabotulinumtoxnA, and 2 underwent sacral neuromodulation Mean old age→ *n* = 13, 1 or 2 medical comorbidities; *n* = 10, 3, or 4 medical comorbidities; *n* = 6 + 5. *n* = 20 W (69%) used anticholinergic treatments before PTNS; *n* = 11 W (38%) used combination therapy during PTNS		Effectiveness and viability of the PTNS technique for the treatment of OAB in elderly patients is observed, being able to choose as a 2nd line treatment. Decrease in the use of anticholinergics by PTNS therapy	Retrospective descriptive study, small sample size which could influence the results. More objective measures should have been used to determine the success of the treatment, such as voiding diary parameters or urodynamic parameters	13/22

PTNS = Percutaneous electrical stimulation of the posterior tibial nerve; PTNS-SEQ = Percutaneous Tibial Nerve Stimultion Service Evaluation Questionnaire; OAB = overactive bladder syndrome; TTNS = Transcutaneous electrical stimulation of the posterior tibial nerve; M = Man; W = Woman; UI = urinary incontinence; OD = overactive detrusor; G = group; m = month; BMI = body mass index; ICIQ-OAB = International Consultation on Incontinence Questionnaire Overactive Bladder Module; ICIQ-LUT = Lower urinary tract symptoms related quality of life questionnaire; UUI = urgency urinary incontinence; USP = Urinary Symptom Profile questionnaire; USP-OAB = Urinary Symptom profile- overactive bladder; OABQ-SF = Overactive Bladder Questionnaire Short Form; OAB-V8 = Overactive Bladder 8 questions Awareness Tool; GIPS = Global Impression of Patient Satisfaction; EMS = electrical stimulation; OAB-q = Short form Symptom Bother; OABSS = Overactive Bladder Symptom Score; UDI-6 = Urinary Distress Inventory, Short Form; IIQ-7 = Incontinence Impact Questionnaire; ICIQ-UI SF = International Consultation on Incontinence Questionnaire; LUTS = Lower Urinary tract Symptoms; SNM = sacral neuromodulation; QoL= Quality of life questionnaire.

**Table 4 healthcare-09-00879-t004:** Characteristics of the systematic reviews and meta-analyses included in the systematic review. Sevilla, ES. 2021.

Author/s	Study Design	Number and Design of Studies	Study Participants	Inerventions	Variables	Results	Conclusions and Limitations
Wibison E., et al., 2015 [28]	Meta-analysis	16 studies 11 RCTs 5 non-comparative prospective studies	Participants with non-neurogenic OAB *n* = 787 480; PTNS 108; antimuscarinics 63 combined therapy 136 sham or placebo treatment More W than M (10/16 W)	PTNS vs sham treatment PTNS vs antimuscarinics PTNS in non-comparative studies	Percentage of responders or patients with positive response Voiding diary parameters (urinary frequency; nocturia; UI and voiding volume)	PTNS vs sham procedure Urinary frequency and UI episodes PTNS vs antimuscarinics PTNS in comparative studies	PTNS therapy is effective for the short-term treatment of OAB, with greater efficacy than with sham treatment, and comparable with antimuscarinic drugs (but with fewer adverse effects). However, multimodal therapy was found to be more effective PTNS could be a maintenance therapy due to its safety and durability Dose, duration, frequency, pulse of PTNS, duration of study follow-up and demographic characteristics of the subjects were highly variable in the studies included
Booth J., et al., 2018 [4]	Systematic review	13 articles 10 RCTs 3 prospective studies	>18 years old with OAB with MUI *n* = 629 → 473 (70%) W and 176 (28%) M, 16 (2%) gender is unknown 36 (18%) →sham treatment 142 (56%) →anticholinergic; 26 (10%)→ pelvic floor and bladder training 9 (4%) → sacral EMS or without treatment	Durability: 4–12 weeks (mean: 7.6 ± 3.6). Total number of sessions 5–90 (mean 21.6 ± 2.3) 30 min/individual session, except 3 of 20 min. 3 studies with daily stimulation, 7 studies with 2 times/week and 2 studies with 1 time/week	Urinary urgency symptoms Urinary frequency Nocturia Number of UI episodes Quality of life Adverse effects Urodynamic changes	Changes in voiding diary Score in OAB symptoms Effectiveness of TTNS Observational studies→ 3/3 General combined result→ 9/13	All studies observed improvements with TTNS treatment. It is safe and tolerable, due to this factor, its low cost, its ease of application and the possibility of self-administration by the patient, more studies are necessary to show its use as a 2nd line treatment
Tutolo M., et al., 2018 [29]	Systematic review	9 articles, all RCTs	Patients with OAB treated by SNM or PTNS	SNM and PTNS PTNS: 4 RCTs→ 388 patients PTNS vs Tolterodine → 3 m (94% W) PTNS vs sham therapy → 3 m PTNS vs placebo → 12 weeks PTNS vs vaginal electrical stimulation→ 12 weeks	Number of UI episodes and severity Number of compress in 24 h Urinary frequency Voiding volume Urinary urgency	PTNS efficacy PTNS safety	There are no high-quality studies able to guide professionals to choose between different treatments. This study shows that sacral stimulation and PTNS are safe and effective. SNM has more long-lasting effects, while PTNS needs to have maintenance treatment Limitations: number of results due to the impossibility of evaluator and patient blinding in the studies
Lo C.W., et al., 2020 [30]	Systematic review and meta-analysis	17 articles, all of them RCTs	Most patients with refractory OAB or patients who have tried 1st or 2nd line of treatment	Treatment with OnabotulinumtoxinA, SNM, and PTNS Heterogeneity between study designs, participants, follow-up, evaluated parameters, OnabotulinumtoxinA dose, and PTNS and SNM protocols	Quality of life Number of UUI episodes in 24 h Urinary frequency in 24 h improvement >50% of symptoms Nocturia Complications	Urinary frequency per day→ 9/17 Number of UI episodes per day→ 7/17 Improvement of 50% or more in symptoms at 12 weeks →8/17	The three modalities are effective and better than placebo for OAB treatment. At 12 weeks, SNM had greater efficacy in UI and urinary frequency, while OnabotulinumtoxinA had more complications Lack of enough data to conduct a meta-analysis in quality of life, urgency, UUI episodes/day, maximum bladder capacity and nocturia 4 studies rated as high risk of bias in the category of ’outcome measurement’ because the self-report results could have been influenced by the placebo effect. Heterogeneity between studies made it difficult to clarify the results
Veeratterapillav R., et al., 2016 [31]	Systematic review	20 studies 6 RCTs 14 controlled clinical trials, prospective cohort studies, and retrospective series	Patients with OAB treated by PTNS and other comparative therapies.	PTNS vs pharmacological treatment PTNS vs placebo PTNS vs sham treatment Follow-up durability varied Different inclusion criteria and definitions of therapy “success” complicated the comparison	Urinary frequency Urinary urgency Nocturia UUI episodes Quality of life Urodynamic study	PTNS urodynamic results and PTNS clinical result PTNS vs anticholinergic therapy PTNS safety and other therapies PTNS cost	PTNS success changed due to informed symptoms by the patient (improvements in frequency and urgency), clinical evaluations (OAB and QoL questionnaires, voiding diary), and observation of aerodynamic variables. Hence, treatment was successful between 54.5% and 79.5% at 12 weeks and 15% and 71% in 1–3 years The studies suggested that PTNS efficacy is better with anticholinergic as a unique treatment, but there was limited evidence of combination therapy efficacy
Wang M., et al., 2020 [32]	Systematic review and meta-analysis	28 articles, 12 RCTs 16 observational studies	246 patients with OAB symptoms treated by PTNS and other comparative therapies	30 min PTNS for 12 weeks in 6/2 studies, the rest of studies had different protocols PTNS vs Tolterodine PTNS vs PTNS + sham treatment PTNS vs TTNS	3-day voiding diary (urinary frequency/day, nocturia/day, number of UUI episodes/day, number of UI episodes, daytime urination frequency/day, voiding volume and urodynamic data) Response rate	Urinary frequency/day → 10/28. Nocturia/día → 13/28 Number of UUI and UI episodes /day → 8/28 and 10/28 Urination frequency /day → 7/28 Voiding volume /day → 8/28 Urodynamic data Treatment response rate → 17/28 Comparison with other therapies	PTNS therapy was shown to be effective and safe for OAB treatment Limitations: heterogeneity of the studies included, however, a subgroup analysis was performed to observe that this factor was due to the study design. Second, evaluation of the improvements and success of the variable was done unconsciously Severe side effects: the most common was pain in the puncture area

RCT = randomized control trial; W = woman; M = man; OD = Overactive detrusor; MA = mean age; G = group; PTNS Percutaneous electrical stimulation of the posterior tibial nerve; TTNS = Transcutaneous electrical stimulation of the posterior tibial nerve; OAB = overactive bladder syndrome; UI = Urinary Incontinence; MUI = mixed urinary incontinence; UUI = Urgency urinary incontinence; EMS = electrical stimulation; SNM = sacral neuromodulation.

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
