# Peer review of "Percutaneous versus Transcutaneous Electrical Stimulation of the Posterior Tibial Nerve in Idiopathic Overactive Bladder Syndrome with Urinary Incontinence in Adults: A Systematic Review"

_healthcare, 2021, doi:10.3390/healthcare9070879_

Round 1

Reviewer 1 Report

 Percutaneous versus Transcutaneous Electrical Stimulation of the Posterior Tibial Nerve in Idiopathic Overactive Bladder Syndrome with urinary incontinence in adults: a Systematic Review 

In this paper, the authors aimed to evaluate the current evidence of TTNS and PTN to treat OAB

This paper will benefit from grammar proofreading by a native English speaker

UI is the involuntary leakage of urine and lack of ability to control urination, accompanied by spontaneous contractions of the detrusor muscle, the contraction of which causes urine to leak out “ — this a very loose definition of urinary incontinence since it ignores UI subtypes. 

In my opinion this a well-structured systematic review with a nice methodology. I think that the manuscript will benefit from grammar proofreading by a native English speaker. 

Author Response

(1) Este artículo se beneficiará de la revisión gramatical realizada por un hablante nativo de inglés.

En primer lugar, muchas gracias por la valoración positiva de nuestro trabajo. 

Nuestro traductor lo ha revisado. (todos estos nuevos cambios han estado en rojo para facilitar el seguimiento)

(2) "La IU es la pérdida involuntaria de orina y la falta de capacidad para controlar la micción, acompañada de contracciones espontáneas del músculo detrusor, cuya contracción hace que la orina se escape": esta es una definición muy vaga de incontinencia urinaria, ya que ignora Subtipos de UI.

El revisor tiene razón al sugerir estas preguntas. Estamos de acuerdo con el revisor en el interés de incluir esta información.

Se ha agregado un párrafo (líneas 53 a 55) para resolver este problema.

(3) En mi opinión, esta es una revisión sistemática bien estructurada con una buena metodología. Creo que el manuscrito se beneficiará de la revisión gramatical de un hablante nativo de inglés.

Gracias por su opinión sobre la metodología y la revisión sistemática estructurada.

Disculpas por este error, el manuscrito fue revisado por un hablante nativo de inglés.

Se ha modificado para solucionar este error. Nuestro traductor ha revisado todo el manuscrito. (todos estos nuevos cambios han estado en rojo para facilitar el seguimiento)

Reviewer 2 Report

Authors should improve introduction, describing treatment options for OAB/UUI and, in general, a scenario in wich the topic of the paper is allocated. Refer to the paper "Managing urinary incontinence in women - a review of new and emerging pharmacotherapy. Bientinesi R, Sacco E.Expert Opin Pharmacother. 2018 Dec;19(18):1989-1997"

Author Response

(1)Authors should improve introduction, describing treatment options for OAB/UUI and, in general, a scenario in wich the topic of the paper is allocated.

 Refer to the paper "Managing urinary incontinence in women - a review of new and emerging pharmacotherapy. Bientinesi R, Sacco E.Expert Opin Pharmacother. 2018 Dec;19(18):1989-1997"

First of all, thank you very much for the positive evaluation of our work.

We agree with the reviewer in the interest of including this information. After reading your reference, and our review, we decided use the reference “[8] Raju, R.; Linder, B.J. Evaluation and treatment of Overactive Bladder in Women. Mayo Clinic Proceedings. Elsevier LTD; 2020, 95, 370-7”. Which was already included in our bibliography, refers to the same treatments and is more up-to-date.

A paragraph (lines 63-67) has been added to solve this issue.
